# New Variants of Squash Mosaic Viruses Detected in Human Fecal Samples

**DOI:** 10.3390/microorganisms9071349

**Published:** 2021-06-22

**Authors:** Fabiola Villanova, Roberta Marcatti, Mayara Bertanhe, Vanessa dos Santos Morais, Flavio Augusto de Padua Milagres, Rafael Brustulin, Emerson Luiz Lima Araújo, Roozbeh Tahmasebi, Steven S. Witkin, Xutao Deng, Eric Delwart, Ester Cerdeira Sabino, Cassio Hamilton Abreu-Junior, Élcio Leal, Antonio Charlys da Costa

**Affiliations:** 1Laboratório de Diversidade Viral, Instituto de Ciências Biológicas, Universidade Federal do Pará, Belem 66075-000, PA, Brazil; fvillanova@gmail.com; 2Departamento de Moléstias Infecciosas e Parasitárias and Instituto de Medicina Tropical da Faculdade de Medicina da Universidade de São Paulo, São Paulo 05403-000, SP, Brazil; robertamarcatti@gmail.com (R.M.); mayarabertanhe@globo.com (M.B.); va.morais@usp.br (V.d.S.M.); roozbeh@usp.br (R.T.); switkin@med.cornell.edu (S.S.W.); sabinoec@gmail.com (E.C.S.); charlysbr@yahoo.com.br (A.C.d.C.); 3School of Veterinary Medicine and Animal Science, University of Sao Paulo, São Paulo 05508-270, SP, Brazil; 4Instituto de Ciências Biológicas, Universidade Federal do Tocantins, Palmas 77001-090, TO, Brazil; flaviomilagres@uft.edu.br (F.A.d.P.M.); eu3rafael@gmail.com (R.B.); 5Public Health Laboratory of Tocantins State (LACEN/TO), Palmas 77016-330, TO, Brazil; 6General Coordination of Public Health, Laboratories of the Strategic Articulation, Department of the Health, Surveillance Secretariat, Ministry of Health (CGLAB/DAEVS/SVS-MS), Brasília 70719-040, DF, Brazil; emerson.araujo@saude.gov.br; 7Department of Obstetrics and Gynecology, Weill Cornell Medicine, 1300 York Avenue, New York, NY 10065, USA; 8Vitalant Research Institute, 270 Masonic Avenue, San Francisco, CA 94143, USA; xdeng@vitalant.org (X.D.); Edelwart@Vitalant.org (E.D.); 9Department Laboratory Medicine, University of California San Francisco, San Francisco, CA 94143, USA; 10Center of Nuclear Energy in Agriculture, Universidade de São Paulo, Piracicaba 3400-970, SP, Brazil; cahabreu@cena.usp.br

**Keywords:** Squash mosaic virus, plant viruses, next generation sequencing, virome, public health

## Abstract

Squash mosaic virus (SqMV) is a phytovirus that infects great diversity of plants worldwide. In Brazil, the SqMV has been identified in the states of Ceará, Maranhão, Piauí, Rio Grande do Norte, and Tocantins. The presence of non-pathogenic viruses in animals, such as phytoviruses, may not be completely risk-free. Similarities in gene repertories between these viruses and viruses that affect animal species have been reported. The present study describes the fully sequenced genomes of SqMV found in human feces, collected in Tocantins, and analyzes the viral profile by metagenomics in the context of diarrhea symptomatology. The complete SqMV genome was obtained in 39 of 253 analyzed samples (15.5%); 97.4% of them belonged to children under 5 years old. There was no evidence that the observed symptoms were related to the presence of SqMV. Of the different virus species detected in these fecal samples, at least 4 (rotavirus, sapovirus, norovirus, parechovirus) are widely known to cause gastrointestinal symptoms. The presence of SqMV nucleic acid in fecal samples is likely due to recent dietary consumption and it is not evidence of viral replication in the human intestinal cells. Identifying the presence of SqMV in human feces and characterization of its genome is a relevant precursor to determining whether and how plant viruses interact with host cells or microorganisms in the human gastrointestinal tract.

## 1. Introduction

One of the greatest challenges in agricultural production is the control of viral diseases. Plant viruses are responsible for decreasing the quantity and quality of crop yield, leading to significant economic losses. Squash mosaic virus (SqMV) is an example of a phytovirus (family *Secoviridae* genus *Comovirus*) that infects melons and a diversity of plants of the family *Cucurbitaceae* [1,2].

SqMV is non-enveloped, 28–30 nm in diameter, and the capsid is icosahedral composed of 60 small proteins (18–26 kDa) and 60 large proteins (37–49 kDa). Genomic RNAs are encapsidated separately into two different types of particles of similar size [3]. The SqMV genome is segmented with bipartite linear ssRNA (positive sense) and is composed of RNA-1 (6–8 kb) and RNA-2 (4–7 kb). Each genomic segment has a VPg linked to its 5′ end and a 3′ poly(A) tract. The 5′- and 3′-UTRs of RNA-1 and RNA-2 are similar in sequence composition. The virion RNA is infectious and serves as both the genome and mRNA.

Viral proteins are usually expressed as large polyproteins, which are cleaved by virus-encoded 3C-like proteinases. Both RNA-1 and RNA-2 segments are translated into two polyproteins, which are further processed to form functional proteins [3]. RNA-1 is translated into a single polyprotein that is processed into five functional proteins: the N-terminal 32K protein (also known as protease co-factor), the 58K protein with sequence motifs characteristic of an NTP-binding helicase, the Vpg, the protease, and the polimerase. The 32K Co-Pro and 58K NTB proteins induce cytopathic structures by the proliferation of ER-derived membranes. The 32K protein also curbs the processing of the RNA-1-encoded polyprotein and affects the processing of the RNA-2-encoded polyprotein [4,5,6]. RNA-2 is translated into three functional proteins: the movement protein (MP), the large capsid protein (LCP), and the small capsid protein (SCP). The MP and the CPs are required for cell-to-cell movement of the virus. The MP is a structural component of tubular structures containing virus-like particles that traverse the cell wall [7,8].

There are two different groups of SqMV, serotypes I and II [9], both of which cause plant diseases worldwide. SqMV is spread mainly by infected seeds, and can be transmitted by chrysomelid beetles, including *Acalymma trivittata* and *Diabrotica* spp. [1,2]. Symptoms and severity of the plant disease vary, depending on many factors, including the strain of the virus, the species and/or variety of plant, and the population of vectors [10]. In Brazil, SqMV has been reported in melons, watermelons, and squash, in the states of Ceará, Maranhão, Piauí, Rio Grande do Norte, and Tocantins [10,11].

Melons and watermelons are commonly consumed by human and animal populations in their raw form, without prior chemical or thermal treatment that inactivates potential pathogenic microorganisms. Among microorganisms that infect these fruits, plant viruses are not considered pathogenic to humans and animals [12].

Some similarities exist between phytoviruses and viruses that affect animal species, including gene repertories [12,13]. While the genes responsible for viral replication and expression may be conserved among plant and animal viruses, genes that determine interactions with their hosts are unique [13]. Indeed, some plant and animal viruses relevant to public health belong to the same viral families, for example, the *Rhabdoviridae* family, which includes Lyssavirus (infecting vertebrates) and Tenuivirus (infecting plants and insects) [12].

Phytoviruses have been identified in mammals, including humans. Their presence in the feces of non-human mammals has been reported [12]. These viruses are stable in the gastrointestinal tract, and, therefore, animal carriers can assume the role of disseminators, contaminating the environment with the virus when defecating [12]. Sequence similarities between viroids and human microDNA were recently identified and pose the need for further investigation on the potential of viroids and their derived small RNAs to cross kingdoms and interact with nucleic acids in humans [14]. There is a case study involving disruption of type III effector-mediated phagocytosis in a human cell line following a plant bacterial infection (*Pseudomonas syringae*) [15]. Studies that identify plant viruses in humans are necessary to better understand their potential to interact with human cells, which can involve modulation of gene expression through interference of their RNA [14,16]. The use of metagenomics analysis is a powerful tool that can contribute to the identification of new viruses relevant to public health.

There is a lack of studies involving the possible presence of phytoviruses in humans and potential health consequences [12]. The purpose of the present study is to describe the completely sequenced genomes of SqMV found in human feces, using samples collected in the state of Tocantins, northern Brazil, through a metagenomics approach. In addition, the profile of all viruses found in samples from individuals with acute gastroenteritis symptoms is delineated.

## 2. Materials and Methods

### 2.1. Population and Specimen Collection

The current cross-sectional surveillance study was performed from 2010 to 2016 in Tocantins (Central region of Brazil). Fecal samples were collected in 38 different localities. A total of 238 stool specimens were obtained from children aged 1–5 years, 3 from children aged 8–15 years, and 7 from adults aged 20–78 years, all with gastroenteritis symptoms. In 5 stool samples the age of the patient was missing. This study was carried out with convenient surveillance specimens, without inclusion or exclusion criteria, and with no characterization of the participants; therefore, epidemiological data (i.e., date of diarrhea onset, vomit episodes or fever) were not available for all patients. In addition, the protocol used failed in identifying possible outbreaks.

### 2.2. Viral Metagenomics

The protocol used to perform deep sequencing was a combination of several protocols applied to viral metagenomics and/or virus discovery according to procedures previously described [17]. Bioinformatics analysis was performed according to the protocol previously described by Deng et al., 2015 [18]. More details about the metagenomic approach used in this study can be found in the Appendix A. The main characteristics of SqMV sequences obtained in this study are summarized in Appendix A. All sequences generated in this study were deposited in the genbank with accession numbers: MZ409700 to MZ409738.

### 2.3. Genetic Analysis

The resulting contigs were subjected to a blast search using Ugene software version 38.1 [19] to identify members of the *Secoviridae* family. Based on the best results (best hits) from the BLASTx search, genomes of SqMV and other related viruses were chosen for further analysis. Next, full or nearly full genomes were aligned using MAFFT software version 7 [20]. To estimate the similarity of sequences, we used a pairwise method implemented in the program SDT version 1.2 [21]. The similarity alignments of every unique pair of sequences were estimated using algorithms implemented in MUSCLE version 3.6 [22]. After the computation of the identity score for each pair of sequences, the program uses the NEIGHBOR component of PHYLIP to compute a tree [23]. The rooted neighbor-joining phylogenetic tree ordered all sequences according to their likely degrees of evolutionary relatedness. Results were presented in a frequency distribution of pairwise identities in a graphical interface. Phylogenetic trees were constructed using the maximum likelihood approach. To obtain reproducible results and provide greater reliability of clustering pattern of trees, the statistical support of branches was evaluated by the approximate likelihood ratio test (aLRT). Trees were inferred using the FastTree version 2.1.11 [24] software and the GTR model plus gamma distribution and the proportion of invariable sites were used. The best model was selected according to the likelihood ratio test (LRT) implemented in the jModeltest version 2.1.10 [25] software.

## 3. Results

### 3.1. Viral Diversity

From the 253 samples analyzed by serology or NGS, the most frequent agents identified were rotavirus (RV) (94.4%), adenovirus (HdV) (84.1%), norovirus (NoV) (74%), astrovirus (HAstV) (52.4%), and sapovirus (SaVs) (23.8%) (the most frequent viruses in our samples are shown in Appendix A). It is important to point out that all patients were affected by acute diarrhea that later resolved after proper medication. A complete description of clinical data and the characterization of other viruses detected in these patients were described in previous studies [26,27,28,29,30,31]. Although SqMV was detected in 51 individuals, full length genomes were recovered in 39 samples. Regarding the 39 patients in which complete genomes of SqMV were recovered, some characteristics are described in Appendix A. One striking finding in these patients was the presence of group A rotavirus, an enteric pathogen, that was not detected by a prior serological analysis. These patients were also positive for other pathogenic viruses, such as norovirus, sapovirus, and adenovirus (see Appendix A).

### 3.2. Phylogenetic Analysis

A detailed phylogenetic analysis was performed to determine the relatedness of the Brazilian SqMV sequences with SqMV detected in other countries. The GenBank IDs of SqMV reference sequences used for the alignment of the segment 1 were AB054688, EU421059, KP223323, MF166756, MF166757, and NC_003799. References used for the segment 2 were AB054689, AF059532, AF059533, EU421060, KP223324, MF166754. MF166755, and NC_003800. The maximum likelihood trees constructed with viral segment 1 (5877nt) and segment 2 (3559nt) indicated that SqMV sequences generated in this study are all monophyletic (Figure 1). We also used a consensus of Brazilian sequences, inferred with nucleotides with a frequency of 60%, to determine the most related SqMV reference strain. We found that in both trees, the Brazilian SqMV consensus was closely related to the SqMV identified in Spain in 2010 (KP223323 for segment 1 and KP223324 for segment 2). 

### 3.3. Genetic Distances and Similarities

We also calculated the genetic distances and similarities of SqMV for two segment sequences obtained in Brazil, with the same sequences from other parts of the world, available in GenBank. The nucleotide distances indicated that the Brazilian SqMV sequences diverged 5% from each other, while SqMV worldwide references diverged 11% in both segments. The nucleotide divergence between Brazilian SqMV and references was 14% in segment 1 and 15% in segment 2 (values indicated in Figure 1). Genetic distances were also calculated in the translated sequences. Table 1 shows the pairwise distances of the translated segment 1 (lower values in the table) and the distances of the translated segment 2 (upper values in the table). For illustrative purposes, only few Brazilian sequences were included in the table. Residues that distinguish the Brazilian sequences from worldwide references are listed in the Appendix A. In addition, nucleotide identity of all pairs of sequences were calculated and showed that SqMV found worldwide have similarities higher than 80% for both genomic segments (Appendix A).

Lower values indicate the genetic distances the translated segment 1. Upper values indicate pairwise distances of the translated segment 2.

## 4. Discussion

Complete SqMV genomes were obtained in 39 of the 253 samples analyzed (15.54%). In total, 97.4% of samples in which we found SqMV belonged to patients under 5 years old. There are previous reports associating phytovirus infection with the appearance of disease symptoms in humans. For example, the tobacco virus (tobacco mosaic virus) has been linked to the appearance of lung cancer in smoking patients in the USA [12,15]. However, until now there is no evidence that plant viruses can infect and replicate in animal cells. In addition, all samples in which SqMV was found were also positive for other viruses. Of the 15 virus species detected in the samples, at least 4 are widely known to cause gastrointestinal symptoms (rotavirus, sapovirus, norovirus, and parechovirus). Additionally, gokushovirus can be found in the human intestinal microbiome [32], and the teno torque virus is commonly present in mixed viral infections [33] This wide variety of gastroenteric viruses found in the analyzed samples waned the possibility of SqMV association with gastroenteritis. The high prevalence of enteric viruses in the state of Tocantins can be related to the low rate of basic sanitation in the area, which contributes not only to the existence of a wide-range occurrence of viruses, but also to increasing the number of human infections [34].

The presence of nucleic acid of phytoviruses in animal feces should be interpreted as being risk free because there is not characteristics indicative of active viral replication. Although unrelated to active replication in human cells [9], they can have other consequences, such as the spread of the viral genomes to new locations. In the current globalized world, people can carry the virus to different parts of the globe. As the viability of these viruses in humans and in the environment is not yet well known, there is no way to predict the consequences of this spread. SqMV sequences reported in this study have likely originated from dietary source.

Evidence for active SqMV replication in the human intestine is still lacking. Further research efforts are needed to determine whether and how plant viruses can interact with intestinal cells or microorganisms in the human gastrointestinal tract. Reporting the presence of SqMV in human feces and making its genome available is a very useful first step for future studies.

## 5. Conclusions

The presence of SqMV in human feces was identified and its complete genome was characterized. Among the limitations of the present study are the narrow geographical distribution of the study subjects and the short collection period. These findings improve our understanding of SqMV classification and its molecular diversity, as well as providing clues to its mechanism of evolution and action in plants, and possible interactions with animals and bacteria.

## Figures and Tables

**Figure 1 microorganisms-09-01349-f001:**
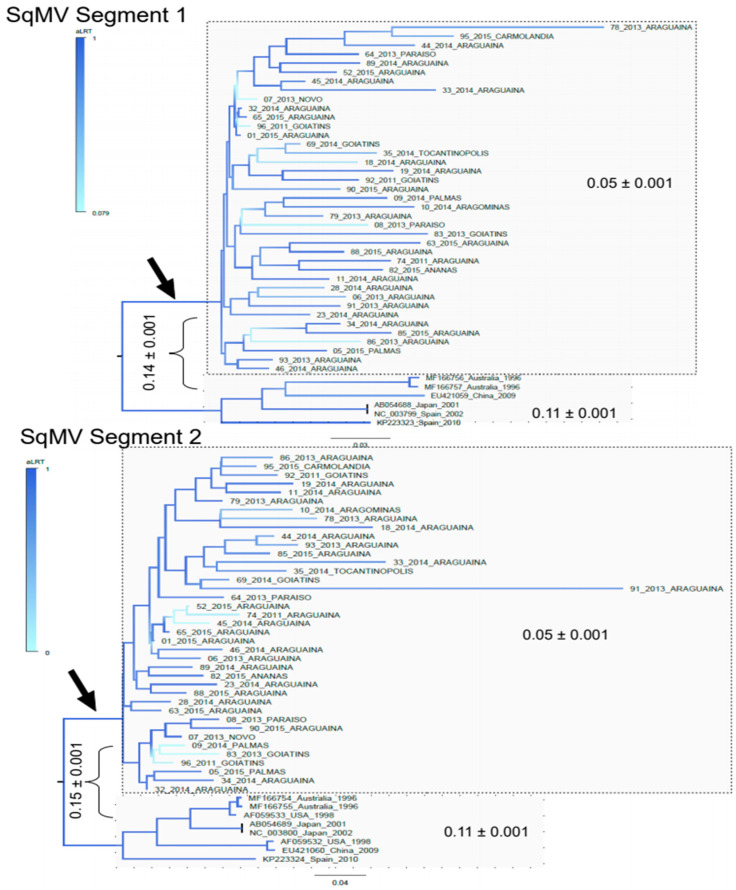
Maximum likelihood tree of genome segments of Squash mosaic viruses. Trees were inferred in FastTree using GTR + gamma correction and the proportion of invariable sites model as selected by the jModelTest software. Branch support was achieved by approximate likelihood ratio test (aLRT) and is shown in a color scale. Scale is indicated in the base of each tree and indicate nucleotide substitutions per site. These trees ware rooted at the midpoint. Numbers indicate the genetic distances and their standard.

**Table 1 microorganisms-09-01349-t001:** Distances of the translated segment 1 and segment 2 of SqMV.

Sequence Name	(1)	(2)	(3)	(4)	(5)	(6)	(7)	(8)	(9)	(10)
(1) 96_2011_GOIATINS		2.20%	1.10%	1.50%	5.29%	5.29%	5.40%	5.40%	6.55%	6.55%
(2) 65_2015_ARAGUAINA	1.14%		1.10%	0.90%	4.36%	4.36%	4.46%	4.46%	5.71%	5.61%
(3) 01_2015_ARAGUAINA	0.98%	1.08%		0.60%	4.46%	4.46%	4.56%	4.56%	5.71%	5.71%
(4) 32_2014_ARAGUAINA	0.81%	0.76%	0.92%		3.94%	3.94%	4.05%	4.05%	5.19%	5.19%
(5) MF166754_Australia_1996	4.98%	6.12%	5.89%	5.72%		0.00%	1.40%	1.40%	3.74%	2.81%
(6) MF166755_Australia_1996	5.09%	6.23%	6.00%	5.83%	0.11%		1.40%	1.40%	3.74%	2.81%
(7) AB054689_Japan_2001	4.47%	5.54%	5.32%	5.15%	2.40%	2.51%		0.00%	3.43%	2.81%
(8) NC_003800_Japan_2002	4.47%	5.54%	5.32%	5.15%	2.40%	2.51%	0.00%		3.43%	2.81%
(9) EU421060_China_2009	5.32%	6.46%	6.23%	6.06%	3.51%	3.62%	3.35%	3.35%		4.67%
(10) KP223324_Spain_2010	4.81%	5.94%	5.72%	5.54%	2.40%	2.51%	1.91%	1.91%	3.68%

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
