# Peer review of "New Variants of Squash Mosaic Viruses Detected in Human Fecal Samples"

_microorganisms, 2021, doi:10.3390/microorganisms9071349_

Round 1

Reviewer 1 Report

In this study Villanova et al. report the identification of genomes of the Squash mosaic virus (SqMV), a phytovirus, in human feces through a metagenomic approach. The dataset analysed here has already been used in four previous publications which were focused on the characterisation of different viruses. The authors describe the presence of SqMV in this dataset and specifically they report SqMV genomes from 39 samples. Given that this is a known plant virus likely introduced through the diet, the study has limited or no public health importance. This descriptive work can be of interest for studying the genetic variability of this virus and, in general, for the characterisation of viruses that are introduced with the diet and may interact with the gut microbiota.

The following points must be addressed before considering it for publication:

The text lacks some basic but important information on this virus such as the type of genome (in this case a single-stranded RNA genome) and its typical genome organisation/gene content (e.g. this can be added as a figure).

The assembled sequences of SqMV are the focus of the study but there’s no mention on important statistics related to these assemblies, e.g. the average depth of coverage of the individual assemblies for each sample, and their contig lengths. These information should be included in a table in the main text or in the supplementary (e.g. using table S1). Beside the average depth of coverage, the authors should at least add one example of the coverage distribution of the reads from a sample on the genome (e.g. with a coverage plot).

The authors do not provide the sequences of the assembled genomes described in the paper. At minimum, the sequences should be made available e.g. as supplementary file, and at least one “representative” sequence must be submitted to genbank and the accession number(s) included in the manuscript. This is obviously necessary before the manuscript can be considered for publication. The Data Availability Statement must be filled accordingly (now there’s no statement).

To clearly show the divergence of the sequences assembled in this study with those already available in public databases, it would be useful to have a table with the pairwise identities (at nucleotide and amino acid levels) with the closest sequences available in genbank. In this regard, in the text there is the following sentence: “The genetic divergence between Brazilian SqMV and references is 14% in segment 1 and 15% in segment 2 (values indicated in Fig. 1).” But figure 1 is missing from the main text.

Minor comments:

The following sentence in the discussion should be clarified/rephrased: “We recognize the possibility that SqMV sequences reported in this study are not infectious and have originated from dietary source.” The most likely hypothesis is that the virus has been introduced with the diet and it is not infectious. Therefore there’s no point in “recognising the possibility” that SqMV is not infectious.

In the abstract, it is not clear what is the exact meaning of the sentence: “.., besides their presence reported in this study may not be autochthonous”. Clarify or consider removing this sentence.

In section 2.2 in the sentence: “were assembled from the sequence readings obtained by de-assembly” it is not clear what the term “de-assembly” means.

The following sentence is vague: “Although low numbers of specimens were included in this study, it does not appear to have been affected the results.” In what sense the results you present here could be affected by this number of samples? The genomes have been assembled from several samples, and the genomes in question are not novel, so there is little room for e.g. misassemblies /chimeric sequences.

The supplementary file has the template of the journal “Viruses”.

Some typos:

“norovirus (NoV) (74%) and astrovirus (HAstV) (52.4%) and sapovirus (SaVs) (23.8%) (the most frequent viruses in our samples are shown in Fig S1.”  ==> “norovirus (NoV) (74%), astrovirus (HAstV) (52.4%) and sapovirus (SaVs) (23.8%) (the most frequent viruses in our samples are shown in Fig S1).”

In the conclusions: “characretized” ==> “characterized”

Author Response

Rev1

In this study Villanova et al. report the identification of genomes of the Squash mosaic virus (SqMV), a phytovirus, in human feces through a metagenomic approach. The dataset analysed here has already been used in four previous publications which were focused on the characterisation of different viruses. The authors describe the presence of SqMV in this dataset and specifically they report SqMV genomes from 39 samples. Given that this is a known plant virus likely introduced through the diet, the study has limited or no public health importance. This descriptive work can be of interest for studying the genetic variability of this virus and, in general, for the characterisation of viruses that are introduced with the diet and may interact with the gut microbiota.

The following points must be addressed before considering it for publication:

The text lacks some basic but important information on this virus such as the type of genome (in this case a single-stranded RNA genome) and its typical genome organisation/gene content (e.g. this can be added as a figure).

Resp: We appreciate this comment. In the new version of the manuscript, a brief paragraph was included to provide this information (lines 66-85).

The assembled sequences of SqMV are the focus of the study but there’s no mention on important statistics related to these assemblies, e.g. the average depth of coverage of the individual assemblies for each sample, and their contig lengths. These information should be included in a table in the main text or in the supplementary (e.g. using table S1). Beside the average depth of coverage, the authors should at least add one example of the coverage distribution of the reads from a sample on the genome (e.g. with a coverage plot).

Resp: For each SqMV we have included information regarding the quality of sequencing (Table S3).

The authors do not provide the sequences of the assembled genomes described in the paper. At minimum, the sequences should be made available e.g. as supplementary file, and at least one “representative” sequence must be submitted to genbank and the accession number(s) included in the manuscript. This is obviously necessary before the manuscript can be considered for publication. The Data Availability Statement must be filled accordingly (now there’s no statement).

Resp: All sequences were submitted to the GenBank and IDs are pending. Soon these sequences will be available.

To clearly show the divergence of the sequences assembled in this study with those already available in public databases, it would be useful to have a table with the pairwise identities (at nucleotide and amino acid levels) with the closest sequences available in genbank. In this regard, in the text there is the following sentence: “The genetic divergence between Brazilian SqMV and references is 14% in segment 1 and 15% in segment 2 (values indicated in Fig. 1).” But figure 1 is missing from the main text.

Resp: The genetic distances of SqMV clades are indicated in the tree (figure 1). Additionally, we have included in this new version of the manuscript a summarized matrix (table 1) showing the genetic distances of translated segment 1 and segment 2 of some sequences. We also described the residues that distinguish Brazilian sequences from SqMV references (table S2). For illustrative purposes, a detailed matrix showing the nucleotide identity of all pairs of sequences was included in figure S2.

Minor comments:

The following sentence in the discussion should be clarified/rephrased: “We recognize the possibility that SqMV sequences reported in this study are not infectious and have originated from dietary source.” The most likely hypothesis is that the virus has been introduced with the diet and it is not infectious.

Therefore there’s no point in “recognising the possibility” that SqMV is not infectious.

Resp: We have changed this sentence to ‘SqMV sequences reported in this study have likely originated from a dietary source.’

In the abstract, it is not clear what is the exact meaning of the sentence: “.., besides their presence reported in this study may not be autochthonous”. Clarify or consider removing this sentence.

Resp: We have changed this sentence to ‘Resp: We have changed this sentence to “The presence of SqMV nucleic acid in fecal samples is likely due to recent dietary consumption and it is not evidence of viral replication in the human intestinal cells.”

In section 2.2 in the sentence: “were assembled from the sequence readings obtained by de-assembly” it is not clear what the term “de-assembly” means.

Resp: We appreciate this comment and have this sentence from the manuscript because details of the next-generation sequencing were included in the supplementary material.

The following sentence is vague: “Although low numbers of specimens were included in this study, it does not appear to have been affected the results.” In what sense the results you present here could be affected by this number of samples? The genomes have been assembled from several samples, and the genomes in question are not novel, so there is little room for e.g. misassemblies /chimeric sequences.

Resp: We appreciate this comment and this sentence has been removed from the new version of the manuscript.

The supplementary file has the template of the journal “Viruses”.

Resp: We did change the format of the manuscript according to the template of Microorganisms

Some typos:

“norovirus (NoV) (74%) and astrovirus (HAstV) (52.4%) and sapovirus (SaVs) (23.8%) (the most frequent viruses in our samples are shown in Fig S1.”  ==> “norovirus (NoV) (74%), astrovirus (HAstV) (52.4%) and sapovirus (SaVs) (23.8%) (the most frequent viruses in our samples are shown in Fig S1).”

Resp: We have changed this sentence accordingly.

In the conclusions: “characretized” ==> “characterized”

Resp: We have corrected this word in the new version of the manuscript.

Reviewer 2 Report

The article analyzes the genomes of Squash mosaic viruses in human feces; viromes were obtained by high-throughput sequencing. Until now, there are few works on the effect of phytoviruses on the human; in this article, the authors tried to shed light on this issue. The complete SqMV genome was obtained in 39 of 253 analyzed samples. The maximum likelihood trees constructed with viral segment 1 and segment 2 indicated that SqMV sequences generated in this study are all monophyletic. The authors acknowledge the possibility that the SqMV sequences reported in this study are not infectious and come from a dietary source. In general, the article is written in clear language and makes a positive impression. But there are a number of minor issues in the text:

Why was not a sample taken from the faeces of a healthy person for control?

I cannot imagine how by identifying any virus in feces (except for those known and capable of doing this), it is possible to identify their pathogenicity for humans. I suppose need to conduct experiments using only the SqMV virus?

An interesting result about the monophyletic of the group of Brazilian SqMVs, which indicates the local spread of this virus, despite the time period of sampling.

For the analysis of nucleotide diversity, it is possible to use the DNAsp program (just a recommendation for the authors).

Secoviridae family - The family should be written in italics.

In several places in the text there are no spaces, perhaps this happened when converting to pdf format.

Among microorganisms that infect these fruits… Maybe use the word «plants» better?

In the text you say that the complete SqMV genome was obtained in 39 of 253 analyzed samples. Why is there no more detailed information on these genomes? How long was it, how many were found?

References - You need to edit the references in accordance with the requirements of the journal. Some words start with an uppercase letter and some with a lowercase letter.

Template with supplementary materials made for Viruses Basel  journal.

The samples described above…  It is necessary to write:  the samples described in this report

Initially, 50mg…  -  Missing space

approximately 300μL - Missing space

…0.45μm…  - Missing space

…Hi-Seq. 2500… - Unnecessary point

What blast analysis program was performed?

I did not see information about the deposited raw data. Have you loaded raw reads into any database?

Figure S1. - For a better perception, it seems to me, you need to indicate the percentage.

Figure S2. What is husavírus? Maybe Squash mosaic viruses?

References - You need to edit the references in accordance with the requirements of the journal.

Author Response

REV2

The article analyzes the genomes of Squash mosaic viruses in human feces; viromes were obtained by high-throughput sequencing. Until now, there are few works on the effect of phytoviruses on the human; in this article, the authors tried to shed light on this issue. The complete SqMV genome was obtained in 39 of 253 analyzed samples. The maximum likelihood trees constructed with viral segment 1 and segment 2 indicated that SqMV sequences generated in this study are all monophyletic. The authors acknowledge the possibility that the SqMV sequences reported in this study are not infectious and come from a dietary source. In general, the article is written in clear language and makes a positive impression. But there are a number of minor issues in the text:

Why was not a sample taken from the faeces of a healthy person for control?

Resp: The study does not include control samples because this is an exploratory survey. Our aim was to determine the diversity and prevalence of viruses in individuals with gastroenteritis from 2010 to 2016 in Tocantins sate Brazil.

I cannot imagine how by identifying any virus in feces (except for those known and capable of doing this), it is possible to identify their pathogenicity for humans. I suppose need to conduct experiments using only the SqMV virus?

Resp: We agree completely with this commentary. The study was conceived to study the repertoire of viruses in individuals with acute gastroenteritis. No attempt was made to link the presence of SqMV with any kind of disease in humans, as has been explained in the manuscript the most probable source of SqMV is dietary intake. It is important to mention that most samples were taken from febrile individuals living in a low-income region with precarious healthy assistance. Besides the recommend medications (paracetamol), one common practice to avoid dehydration in that region is the consumption of fruits (such as melons and watermelons). It is plausible that the recent consumption of these fruits by some individuals have affected the amount of viral sequences during the next-generation sequencing.

An interesting result about the monophyletic of the group of Brazilian SqMVs, which indicates the local spread of this virus, despite the time period of sampling.

Resp: We think that SqMV is endemic in the region of this study. Besides sequences detected in our samples likely represent a local SqMV variant.

For the analysis of nucleotide diversity, it is possible to use the DNAsp program (just a recommendation for the authors).

We also appreciate the recommendation of using DNAsp, this is an old-school classic software.

Secoviridae family - The family should be written in italics.

Resp: We have corrected this in the new version of the manuscript.

In several places in the text there are no spaces, perhaps this happened when converting to pdf format.

Among microorganisms that infect these fruits… Maybe use the word «plants» better?

Resp: ‘fruits’ in this sentence refer to ‘melons and watermelons’ mentioned in the beginning of the paragraph

In the text you say that the complete SqMV genome was obtained in 39 of 253 analyzed samples. Why is there no more detailed information on these genomes? How long was it, how many were found?

Resp: We have included a table (Table S3) to summarize the main features of each sequence of SqMV.

References - You need to edit the references in accordance with the requirements of the journal. Some words start with an uppercase letter and some with a lowercase letter.

Resp: References were formatted accordingly.

Template with supplementary materials made for Viruses Basel  journal.

Resp: We have changed the supplementary material according to the Microorganisms format

The samples described above…  It is necessary to write:  the samples described in this report

Resp: Thanks for pointing this in the methodology section. We have changed this accordingly.

Initially, 50mg…  -  Missing space

approximately 300μL - Missing space

…0.45μm…  - Missing space

…Hi-Seq. 2500… - Unnecessary point

Resp: We appreciate you for pointing out these things in the manuscript.

What blast analysis program was performed?

Resp: We have used a pipeline (describe in the methodology) that uses blast- based search. Besides, we also have used UGENE that uses blast in bed.

I did not see information about the deposited raw data. Have you loaded raw reads into any database?

Resp: All sequences have been deposited in Genbank and IDs are pending.

Figure S1. - For a better perception, it seems to me, you need to indicate the percentage.

Resp: We have changed this figure and the new manuscript to show values in percentage.

Figure S2. What is husavírus? Maybe Squash mosaic viruses?

Resp: Our fault this has been corrected in the new manuscript

References - You need to edit the references in accordance with the requirements of the journal.

Resp: We have changed the references following the format of Microorganisms journal.

Round 2

Reviewer 1 Report

I’m satisfied with the revisions made and I recommend it for publication.

I suggest to have an additional check on the main text as there seems to be some typos, e.g. in the legend of figure 1.

The text of figure 1 is not readable, so make sure a higher quality image is used, or increase the text size.